# Severe COVID-19 in pregnancy has a distinct serum profile, including greater complement activation and dysregulation of serum lipids

**Marie Altendahl**[1], **Thalia Mok**[1], **Christine Jang**[1], **Seungjun Yeo**[2], **Austin Quach**[2], **Yalda Afshar**[1] *

**1** Division of Maternal Fetal Medicine, Department of Obstetrics and Gynecology, David Geffen School of Medicine at UCLA, Los Angeles, California, United States of America, **2** Dalton Bioanalytics Inc., Los Angeles, California, United States of America

* yafshar@mednet.ucla.edu

**Data Availability Statement:** All relevant data are within the manuscript and its Supporting Information files.

## Abstract

### Background

Pregnancies complicated by Coronavirus Disease 2019 (COVID-19) are at an increased risk of severe morbidity due to physiologic changes in immunologic, cardiovascular, and respiratory function. There is little is known about how severity of COVID-19 changes protein and metabolite expression in pregnancy.

### Objective

This study aims to investigate the pathophysiology behind various clinical trajectories in pregnant patients diagnosed with COVID-19 using multi-omics profiling.

### Study design

This is a prospective cohort study of 30 pregnant patients at a single tertiary care center. Participants were categorized by severity of COVID-19 disease (control, asymptomatic, mild/moderate, or severe). Maternal serum samples underwent LC-MS-based multiomics analysis for profiling of proteins, lipids, electrolytes, and metabolites. Linear regression models were used to assess how disease severity related to analyte levels. Reactome pathway enrichment analysis was conducted on differential analytes.

### Results

Of 30 participants, 25 had confirmed diagnosis of COVID-19 (6 asymptomatic (one post-infection), 13 mild/moderate (all post-infection), 6 severe), and 5 participants were controls. Severe COVID-19 was associated with distinct profiles demonstrating significant proteomic and lipidomic signatures which were enriched for annotations related to complement and antibody activity. (FDR < 0.05). Downregulated analytes were not significantly enriched but consisted of annotation terms related to lipoprotein activity (FDR > 0.2). Post-infection mild/moderate COVID-19 did not have significantly altered serum protein, metabolite, or lipid metabolite levels compared to controls.

**Funding:** Funding by Carolyn L. Kuckein Student Research Fellowship to M.A. and National Institute of Child Health and Human Development, K12 HD000849 to Y.A. Funding sources were not involved in study design, data collection, analysis and interpretation, writing of manuscript, or the decision to submit the findings of this study for publication.

**Competing interests:** The study was approved by the Institutional Review Board (UCLA IRB# 20-000579). After patient was identified, participants underwent screening and eligible participants were contacted by a study coordinator by phone or email. Verbal or email consented was obtained. All co-authors have read and approved the manuscript in its current submitted form. Authors AQ and SY are co-founders of, are employed by, and own stock in Dalton Bioanalytics Inc. This does not alter our adherence to PLOS ONE policies on sharing data and materials. All other authors report no conflict of interest.

## Conclusions

Pregnancies with severe COVID-19 demonstrate greater inflammation and complement activation and dysregulation of serum lipids. This altered multiomic expression provides insight into the pathophysiology of severe COVID-19 in pregnancy and may serve as potential indicators for adverse pregnancy outcomes.

## Introduction

In December 2019, severe acute respiratory syndrome coronavirus 2 (SARS-CoV-2) began spreading, rapidly leading to a global pandemic [1, 2]. Pregnant persons are considered a high-risk population during infectious disease outbreaks as they have increased susceptibility to infections and its sequelae due to the physiological changes of pregnancy [2, 3]. The clinical presentation of Coronavirus Disease 2019 (COVID-19) in pregnancy is variable and can range in severity from asymptomatic to critical illness with acute respiratory distress syndrome (ARDS), multiorgan failure, and in some cases, death [1, 2, 4]. It has been demonstrated that pregnancy is associated with an increased risk for more severe COVID-19 disease, requiring ICU admission or mechanical ventilation [1, 2, 4]. Though it is unclear what leads to the differentiation of disease severity in pregnant persons with SARS-CoV-2 infection, we hypothesize that greater inflammation and immunologic dysfunction contribute to more severe disease.

Multi-omics research is uncovering distinct metabolic and proteomic serum compositions based on COVID-19 severity and elucidating mechanisms driving immunologic dysfunction seen in severe COVID-19 [5, 6]. Su et al. found that moderate and severe COVID-19 cases have significantly elevated inflammation and a large decrease in blood nutrients compared to those with mild disease [6]. Investigating protein and metabolite expression in COVID-19 using multi-omics provides insight into the pathophysiology of COVID-19 and into biochemical parameters with the potential to identify patients at greatest risk for developing COVID-19 complications.

Little is known about how the severity of COVID-19 changes biochemical expression in the serum of pregnant people. Alterations in protein expression during pregnancy may increase the risk for development of increased COVID-19 severity. This study aims to investigate the pathophysiology behind various clinical trajectories in pregnant patients diagnosed with COVID-19 by using multiomics profiling. Investigation of multiomic expression in pregnant participants with COVID-19 will further our understanding of the pathophysiology of COVID-19 in pregnancy and may serve to elucidate potential indicators for adverse pregnancy outcomes.

## Materials and methods

Pregnant people with COVID-19 infection were actively enrolled at the University of California, Los Angeles between October 1st, 2020 and February 28th, 2021 through a prospective cohort study evaluating maternal and neonatal outcomes of pregnancies with COVID-19 infection. The study was approved by the Institutional Review Board (UCLA IRB# 20–000579). After patient was identified, participants underwent screening and eligible participants were contacted by a study coordinator by phone or email. Verbal or email consented was obtained. Confirmed COVID-19 infection was defined as being SARS-CoV-2 positive by nasopharyngeal RT-PCR. Eligibility for the study included participants >13 years old,

pregnant at the time of enrollment, with necessary clinical data and biospecimen collection during pregnancy. Healthy pregnant controls without COVID-19 infection, defined as a negative SARS-CoV-2 positive nasopharyngeal RT-PCR test, were concurrently recruited.

## Demographic and clinical data

Baseline demographic, clinical data, and clinical labs (ie. IL-6, ferritin, D-dimer) were collected via electronic medical record review. COVID-19 severity was categorized as asymptomatic, mild, moderate, or severe based on the NIH guidelines [7]. Asymptomatic illness is defined as patients who tested positive for COVID-19 but have no symptoms. Mild illness includes individuals who have symptoms of COVID-19, but do not have shortness of breath or abnormal chest imaging. Moderate illness is defined as individuals who have symptoms or imaging consistent with lower respiratory disease, but oxygen saturations remain ≥94% on room air. Severe illness is defined as SpO2 <94% on room air, PaO2/FiO2 <300 mm Hg, a respiratory rate >30 breaths/min, or lung infiltrates >50% or any individual with respiratory failure, septic shock, or multiple organ dysfunction secondary to COVID-19 [8].

## Biospecimen collection and processing

Peripheral blood specimens were obtained from consented patients at the time of study enrollment, as close to diagnosis of acute COVID-19 infection as possible. 2-3mL of blood were drawn from participants and collected in a red top tube without any anticoagulants and preservatives. The tube was centrifuged and spun at 1500rpm for 10 minutes in room temperature. Using a pipette, aliquots of 500uL of serum were drawn out of tube without disrupting the bottom red blood cell layer. The cryovial tubes were labeled with corresponding IDs and time of collection to store in biobank. The serum cryovials were stored in the -80-degree freezer for analyses. Of the 267 available samples, 31 serum samples were chosen based on COVID-19 severity (control, asymptomatic, mild/moderate, and severe) for analyses. Each sample was from a unique participant.

## LC-MS biochemical analysis

Protein, lipid, and small molecule multiomic analysis of serum samples was performed by a commercial research laboratory (Dalton Bioanalytics Inc., Los Angeles, CA). Briefly, after randomization 100 microliters of each serum sample were spiked with internal standards, denatured in an ammonium bicarbonate buffered methanol solution, digested with trypsin, precipitated with ethanol and acetonitrile, and clarified by high-speed centrifugation. The extracted supernatant was analyzed on a liquid chromatography mass spectrometry instrument. Mixed mode RP-HILIC liquid chromatography and high-resolution mass spectrometry was employed for relative quantification of biochemicals in the sample preparations. MS1 data was collected for quantification in both positive and negative ion modes. Ions were identified via matching of data dependent MS2 spectra to proteomic, lipidomic, and metabolomic mass spectral libraries, and filtered on conventional match score thresholds. Identifications were used for calibration of MS1 data and for label free quantification of peak intensities by matching identifications between runs (MBR) based on m/z and retention time using in-house software.

## Statistical analysis

Serum samples underwent multiomic profiling however a single outlier sample was omitted from further analysis likely due to mis-injection (technical error). Relative intensities were

log10 transformed and statistically corrected for the technical effects run order and extraction efficiency (average internal standard intensity). The peak areas for multiple analyte ionic forms were averaged into molecule-level relative quantities (e.g. peptides into a protein, lipid species into lipid isomers, etc.). These molecule-wise relative quantities were used for downstream data analysis.

Participant demographic and clinical characteristics were clustered based on pairwise correlation distance to identify potentially important associations. The serum multiome was subsequently screened for differential associations by COVID-19 severity using unadjusted and adjusted linear regression models (age, race, and gestational age at COVID-19 diagnosis / draw). Proteins nominally associated with severe COVID-19 ($p < 0.05$) were tested for enrichment of functional annotations using DAVID enrichment analysis [9]. Kolmogorov–Smirnov testing was used to test for differential enrichment of specific lipid classes.

## Results

Of the 31 serum samples collected, 30 samples were included in the analysis, as one sample was identified as an outlier and excluded from analysis. 25 participants were SARS-CoV-2 positive and five were gestational-age matched healthy pregnant participants serving as controls. Of those with COVID-19 infection, 6 were asymptomatic, 13 had mild or moderate severity, and 6 had severe disease. Table 1 describes the demographics and clinical characteristics of all participants. 2 participants did not deliver within the University of California, Los Angeles Health system and thus were missing birthing outcome variables such as gestational age at delivery. The average maternal age of participants was 32.7 years old (19–41 years, SD 5.01). Participants with severe COVID-19 were more likely to be Hispanic ($p = 0.018$) and obese ($p = 0.011$). Besides obesity, there were no significant differences in presence of maternal comorbidities by COVID-19 severity or in controls ($p > 0.05$). With respect to clinical characteristics of COVID-19 infection, there were no significant differences in trimester of infection ($p = 0.06$). 2 participants were diagnosed in the first trimester, 13 in the second trimester, and 11 in the third trimester. Of the 19 participants with symptomatic COVID-19, the most reported symptoms were cough (10/19, 52.6%), shortness of breath (9/19, 47.4%), myalgias (9/19, 47.4%), and nasal congestion (8/19, 42.1%). Those with mild or moderate disease were more likely to endorse headache, anosmia/ageusia, and nasal congestion, whereas those with severe disease were more likely to endorse shortness of breath. Of the participants with severe illness, 6 required O2, 2 were intubated, 2 developed ARDS, 1 required extra corporeal membrane oxygenation (ECMO). Additional clinical characteristics of participants with severe COVID-19 are described in Table 2. No participants died secondary to complications from COVID-19.

Precipitous changes in maternal serum were seen in those with severe COVID-19 infection in pregnancy. Principal components analysis shows a clear separation of severe cases versus all others (Fig 1A). Of the 496 serum proteins measured, 87 proteins were significantly associated with severe COVID-19, with 40 increased and 47 decreased in severe infections (FDR < 0.05, Fig 1B, Table 3, S1 Table). Of the 467 measured lipids, 136 lipids were associated with severe COVID-19, with 51 increased and 85 decreased (FDR < 0.05, Table 3). Upregulated lipid classes were enriched for diacylglycerols (DG), triacylglycerols (TG), fatty acids (FA), and phosphatidylethanolamines (PE), whereas downregulated lipids were enriched for phosphatidylcholine classes such as phosphatidylcholines (PC), lysophosphatidylcholines (LysoPC), plasmenyl phosphatidylcholines (Plasmenyl-PC), plasmanyl phosphatidylcholines (Plasmanyl-PC), plasmenyl phosphatidylethanolamines (Plasmenyl-PE), (FDR < 0.05, Table 3). Of the 122 metabolites/compounds measured, 19 were associated with severe

**Table 1. Demographic and clinical characteristics categorized by severity of COVID-19 infection.**

| Characteristic | N | Asymptomatic (n = 6) | Mild/Moderate (n = 13) | Severe (n = 6) | Control (n = 5) | p-value |
|---|---|---|---|---|---|---|
| Maternal age | 30 | 31.8 ± 7.1 | 34.3 ± 3.2 | 29.5 ± 5.4 | 33.8 ± 5.2 | 0.250 |
| Mean ± SD (Range) | | (22–41) | (28–39) | (19–33) | (28–40) | |
| Gestational age at delivery | 28 | 39.6 ± 0.9 | 39.3 ± 0.7 | 37.1 ± 5.3 | 38 ± 2.3 | 0.228 |
| Mean ± SD (Range) | | (38.1–40.6) | (37.3–40.3) | (29.4–41.4) | (34.3–39.9) | |
| *Race/Ethnicity*, n (%) | 30 | | | | | 0.018* |
| Asian | | 0 (0.0) | 3 (23.1) | 0 (0.0) | 1 (20.0) | |
| Black | | 1 (16.7) | 0 (0.0) | 0 (0.0) | 0 (0.0) | |
| White | | 1 (16.7) | 5 (38.5) | 1 (16.7) | 4 (80.0) | |
| Hispanic | | 2 (33.3) | 1 (7.7) | 5 (83.3) | 0 (0.0) | |
| Other | | 2 (33.3) | 4 (30.8) | 0 (0.0) | 0 (0.0) | |
| *Maternal Comorbidities*, n (%) | | | | | | |
| Obesity | 30 | 0 (0.0) | 2 (15.4) | 4 (66.7) | 0 (0.0) | 0.011* |
| Chronic HTN | 30 | 0 (0.0) | 0 (0.0) | 0 (0.0) | 0 (0.0) | *N/A* |
| Asthma | 30 | 0 (0.0) | 1 (7.7) | 1 (16.7) | 0 (0.0) | 0.621 |
| Anemia | 30 | 1 (16.7) | 1 (7.7) | 2 (33.3) | 1 (20.0) | 0.574 |
| Thyroid dysfunction | 30 | 1 (16.7) | 1 (7.7) | 1 (16.7) | 1 (20.0) | 0.881 |
| Infection other than COVID-19 | 30 | 3 (50.0) | 5 (38.5) | 3 (50.0) | 0 (0.0) | 0.280 |
| *Obstetric Outcomes*, n (%) | | | | | | |
| Gestational HTN | 30 | 0 (0.0) | 3 (23.1) | 2 (33.3) | 1 (20.0) | 0.523 |
| Pre-eclampsia | 30 | 0 (0.0) | 2 (15.4) | 0 (0.0) | 0 (0.0) | 0.423 |
| Gestational diabetes | 30 | 0 (0.0) | 3 (23.1) | 2 (33.3) | 0 (0.0) | 0.286 |
| *Trimester of Infection*, n (%) | 25 | | | | | 0.062 |
| First trimester | | 0 (0.0) | 2 (15.4) | 0 (0.0) | *N/A* | |
| Second trimester | | 1 (16.7) | 9 (69.2) | 3 (50.0) | *N/A* | |
| Third trimester | | 5 (83.3) | 2 (15.4) | 3 (50.0) | *N/A* | |
| *COVID-19 Symptoms*, n (%) | | | | | | |
| Cough | 25 | 0 (0.0) | 5 (38.5) | 5 (83.3) | *N/A* | 0.013* |
| Fever | 25 | 0 (0.0) | 1 (7.7) | 2 (33.3) | *N/A* | 0.163 |
| Shortness of breath | 25 | 0 (0.0) | 3 (23.1) | 6 (100.0) | *N/A* | 0.001** |
| Myalgia | 25 | 0 (0.0) | 5 (38.5) | 4 (66.7) | *N/A* | 0.053 |
| Headache | 25 | 0 (0.0) | 6 (46.2) | 0 (0.0) | *N/A* | 0.026* |
| Anosmia/ageusia | 25 | 0 (0.0) | 3 (23.1) | 0 (0.0) | *N/A* | 0.207 |
| Nasal congestion | 25 | 0 (0.0) | 8 (61.5) | 0 (0.0) | N/A | 0.004** |

Infection other than COVID-19 included: HIV, hepatitis C, herpes simplex virus, chlamydia trachomatis, allergic bronchopulmonary aspergillosis, group B strep and diagnosis of urinary tract infection or chorioamnionitis. Two participants did not deliver at UCLA, thus gestational age at delivery was only available for twenty-eight participants.

*p-value <0.05

**p-value <0.01.

COVID-19, with 10 increased and 9 decreased (FDR < 0.05, Table 3). The most significant associations are listed in Table 3 (full data, S1 Table). A joint metabolite-protein Reactome pathway enrichment analysis revealed that upregulated analytes were enriched for annotations related to 'Complement cascade', 'Signaling by the B Cell Receptor (BCR)', 'Fc epsilon receptor (FCERI) signaling', and 'FCGR activation' (FDR < 0.05, Fig 2, S2 Table). Downregulated analytes were not significantly enriched for annotations but consisted of annotation terms such as 'Plasma lipoprotein assembly, remodeling, and clearance' (FDR = 0.30).

**Table 2. Demographic and clinical characteristics of cohort with severe COVID-19 (n = 6).**

| Characteristic | Patient 1 | Patient 2 | Patient 3 | Patient 4 | Patient 5 | Patient 6 |
|---|---|---|---|---|---|---|
| Maternal age | 33 | 19 | 33 | 29 | 33 | 30 |
| Gestational age (weeks) at delivery | N/A | 41.4 | 37.9 | 39.6 | N/A | 29.4 |
| *Maternal Comorbidities* | | | | | | |
| Obesity | 0 | 1 | 1 | 1 | 0 | 1 |
| Chronic HTN | 0 | 0 | 0 | 0 | 0 | 0 |
| Asthma | 0 | 0 | 1 | 0 | 0 | 0 |
| Anemia | 0 | 0 | 0 | 0 | 1 | 1 |
| Thyroid dysfunction | 0 | 0 | 0 | 0 | 0 | 1 |
| Infection other than COVID-19 | 1 | 0 | 1 | 1 | 0 | 0 |
| *Obstetric Outcomes* | | | | | | |
| Gestational HTN | 0 | 1 | 1 | 0 | 0 | 0 |
| Pre-eclampsia | 0 | 0 | 0 | 0 | 0 | 0 |
| Gestational diabetes | 1 | 0 | 1 | 0 | 0 | 0 |
| Trimester of COVID-19 Infection | 2 | 3 | 2 | 3 | 3 | 2 |
| *COVID-19 Symptoms* | | | | | | |
| Cough | 1 | 1 | 1 | 1 | 1 | 0 |
| Fever | 0 | 0 | 0 | 1 | 1 | 0 |
| Shortness of breath | 1 | 1 | 1 | 1 | 1 | 1 |
| Myalgia | 1 | 0 | 1 | 1 | 1 | 0 |
| Headache | 0 | 0 | 0 | 0 | 0 | 0 |
| Anosmia/ageusia | 0 | 0 | 0 | 0 | 0 | 0 |
| Nasal congestion | 0 | 0 | 0 | 0 | 0 | 0 |
| COVID-19 Complications | | | | | | |
| O2 Saturation <94% | 1 | 1 | 1 | 1 | 1 | 1 |
| Supplemental O2 Required | 1 | 1 | 1 | 1 | 1 | 1 |
| Intubation | 1 | 0 | 0 | 0 | 0 | 1 |
| ARDS | 0 | 0 | 0 | 1 | 0 | 1 |
| ICU Admission | 1 | 1 | 0 | 1 | 0 | 1 |
| ECMO | 0 | 0 | 0 | 0 | 0 | 1 |
| Inflammatory Markers | | | | | | |
| IL-6 | N/A | 2.2 | N/A | 3.9 | N/A | 53.5 |
| D-dimer | 0.81 | 0.84 | 0.39 | 1.06 | 0.84 | 2.04 |
| Ferritin | 584 | 77 | 76 | 250 | 63 | 145 |

Post-infection mild/moderate COVID-19 patients did not have significantly altered serum protein or compound levels compared to controls. We found significant associations with COVID-19 severity however these were largely driven by the severe cases—when excluding the severe cases, there were much fewer significant associations, however the consistency associations before and after exclusion of severe cases was moderately correlated (cor = 0.48). The addition of the adjustment variables maternal age, race, obesity, and age at gestation to the severe COVID-19 models reduced the strength of their associations but remained concordant with unadjusted model (cor>0.87). These findings support that severe COVID-19 in pregnancy is associated with substantial changes in proteomic and lipidomic serum expression, distinguishing them from those with asymptomatic and mild/moderate disease.

Considering the limited statistical power of our study (severe cases n = 6 of 30), we sought independent external validation. We compared our findings to the results of Overmyer et al. another multiomic study of COVID-19 to assess agreement between pregnant mothers and

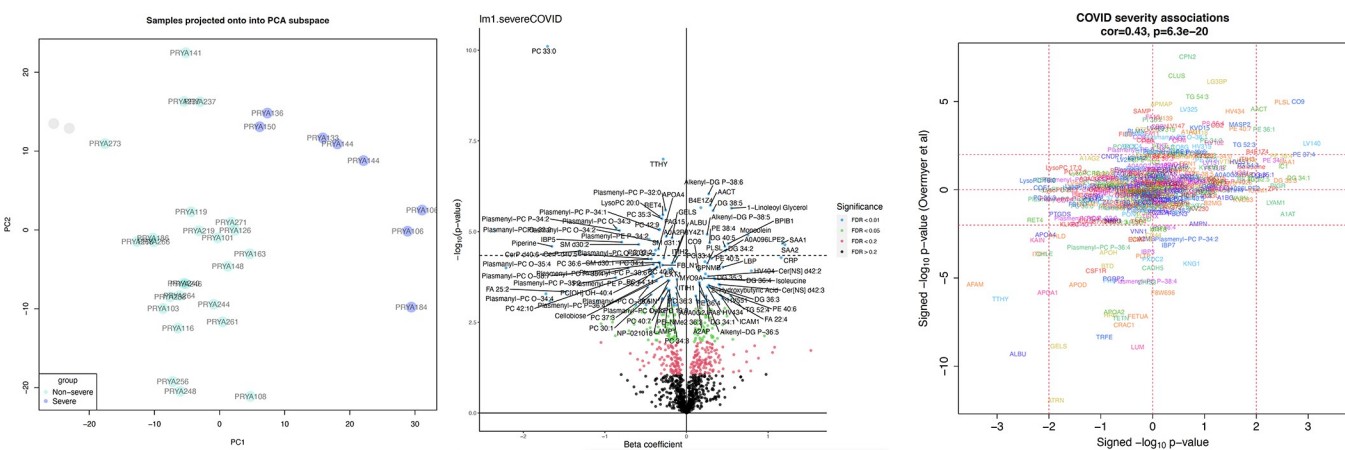

**Fig 1.** a. Principal components analysis showing clear separation of severe COVID-19 cases versus all others. b. Volcano plot illustrating proteins altered in severe SARS-CoV-2 positive pregnancies. c. COVID-19 severity associations.

severe COVID-19 in the general population [5]. Taking the intersecting biomolecules between our studies, we find a moderate agreement between our COVID-19 severity associations and theirs (cor = 0.43, p = 6e-20, S1 Fig). Top concordant hits include the downregulated Albumin and upregulated Complement 9.

Rather than building a predictive model from biochemicals measurable using specialized LC-MS instrumentation we instead sought to assess the viability of utilizing clinical parameters to build a predictor of severe COVID-19 supported by our study. By using a signed simple average of scaled Albumin (general liver marker), C-reactive protein (acute phase

**Table 3. Top proteins and metabolites with the greatest associations in pregnant participants with severe COVID-19.**

| Analyte | Type | Beta | FDR |
|---|---|---:|---:|
| PC 16:0_17:0 | lipid | -1.70 | 8.6E-08 |
| Transthyretin | protein | -0.29 | 5.5E-05 |
| Alkenyl-DG P-18:1_20:5 | lipid | 0.27 | 3.3E-04 |
| Alpha-1-antichymotrypsin | protein | 0.29 | 3.6E-04 |
| C3/C5 convertase | protein | 0.18 | 3.6E-04 |
| 1-Linoleoyl Glycerol | compound | 0.55 | 3.6E-04 |
| Plasmenyl-PC P-16:0_16:0 | lipid | -0.26 | 3.6E-04 |
| DG 18:2_20:3 | lipid | 0.33 | 3.6E-04 |
| Retinol-binding protein 4 | protein | -0.29 | 4.2E-04 |
| LysoPC 20:0 | lipid | -0.33 | 4.2E-04 |
| PC 15:1_20:2 | lipid | -0.30 | 4.4E-04 |
| Plasmanyl-PC O-16:0_18:3 | lipid | -0.86 | 6.4E-04 |
| PC 20:4_22:5 | lipid | -0.33 | 6.4E-04 |
| Plasmenyl-PC P-20:0_14:1 | lipid | -0.34 | 6.4E-04 |
| Plasmanyl-PC O-20:0_12:2 | lipid | -2.34 | 6.4E-04 |
| Plasmenyl-PC P-16:0_18:2 | lipid | -0.82 | 6.4E-04 |
| Alkenyl-DG P-18:0_20:5 | lipid | 0.25 | 7.6E-04 |
| Monoolein | compound | 0.75 | 7.6E-04 |
| Phospholipase A2 group XV | protein | -0.25 | 7.8E-04 |
| Gelsolin | protein | -0.18 | 8.2E-04 |

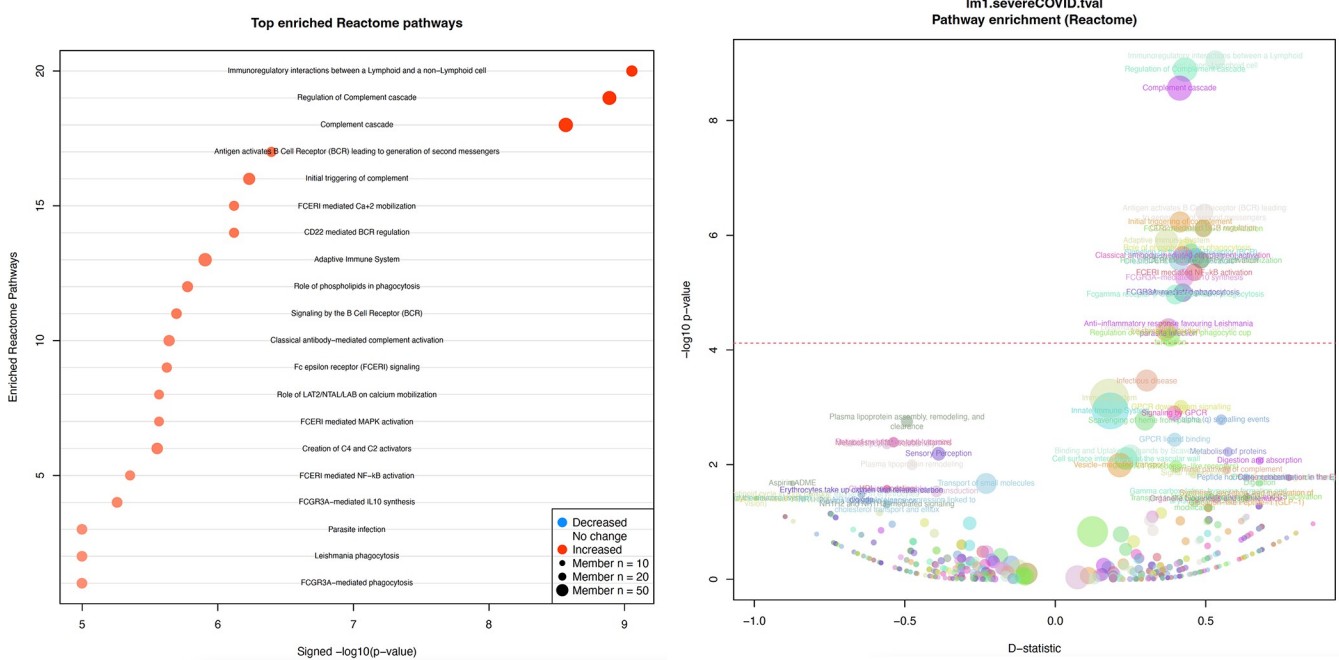

**Fig 2.** a. Top enriched Reactome pathways. b. Severe COVID-19 pathway enrichment.

inflammatory marker), and Apolipoprotein A1 (major HDL particle component), this composite measure performed better than the individual markers (AUC for Albumin = 92%, APOA1 = 91%, CRP = 92%, composite = 98%). Independent follow-up using clinical absolute measurements of Albumin, HDLC, and CRP should verify this finding and evaluate whether these markers can serve as a predictive prognostic marker of severe COVID-19 infection prior to patient progression.

## Comment

**Principle findings.** The purpose of this study was to identify metabolic and proteomic differences seen in pregnancies with and without COVID-19 and evaluate for metabolic and proteomic differences between severity of COVID-19 infection. Our findings demonstrated that pregnant persons with post-infection mild/moderate COVID-19 infection had multiomic profiles like controls, but those with asymptomatic infection and more so severe disease demonstrated distinct changes in metabolite and protein expression. We identified the biochemicals that were significantly upregulated/downregulated in pregnant participants with severe COVID-19 compared to those with less severe COVID-19. Among the upregulated biochemicals were acute phase response proteins, serine protease inhibitors (SERPINs), complement factors, and acylglycerols. Of downregulated biochemicals, a subset was related to lipid metabolism such as apolipoproteins and phosphatidylcholines. These findings emphasize that pregnancies with severe COVID-19 exhibit greater inflammation and complement activation, with altered lipid metabolism.

**Results in the context of what is known.** COVID-19 severity and death secondary to COVID-19 is strongly associated with uncontrolled immune responses, termed "cytokine storm" [10, 11]. Upon entry into respiratory epithelia via the ACE2 receptor, SARS-COV-2 triggers a proinflammatory response through pathogenic Th1 cells and interferons [10, 11]. In severe cases, massive release of inflammatory cytokines results in lymphopenia, thrombosis,

and mononuclear cell infiltration in various tissues throughout the body [10–12]. Non-pregnant patients with severe COVID-19 have been found to have greater levels of IL-2, IL-6, IL-7, IL-10, IP-10, MCP-1, TNF-α, macrophage inflammatory protein 1 alpha, and granulocyte-CSF than those with mild and moderate infections [11, 13]. The observation of a marked inflammatory response in severe COVID-19 infection has also been described in pregnancy [14]. Our study demonstrates similar findings with pregnancies with severe COVID-19 displaying proteomic signature and altered metabolites associated with increased inflammation. This further supports the hypothesis that the dysregulated immune response following SARS-CoV-2 infection plays a primary role in development of severe disease, both in the pregnant and non-pregnant population.

An important driver of inflammation is the complement system, which initiates phagocytosis, chemotaxis, leukocyte activation, and release of inflammatory mediators [15]. Zinellu & Mangoni found that C3 and C4 concentrations were significantly decreased, indicating increased complement activation, in patients with more severe COVID-19 in the non-pregnant population and that increased complement activation was significantly associated with greater mortality [15]. Additionally, a proteomic study correlating expression with COVID-19 severity demonstrated that out of 93 differentially expressed proteins 50 belonged to activation of the complement system, platelet degranulation, and macrophage function [16]. We similarly showed that pregnant participants with severe COVID-19 had upregulation of C3/C5 convertase, complement component C2, complement component C9, and complement C1q subcomponent subunit C, indicating increased complement activation. Our findings suggest that severe COVID-19 disease states in pregnancy is characterized by complement activation, like that of the non-pregnant population. The upregulation of complement activation may contribute to the vulnerability pregnant patients have for disease complications as complement activation in pregnancy is associated with poorer maternal outcomes [17, 18]. Measurement of complement activation in pregnancies affected by COVID-19 may indicate disease severity and possibly identify patients at greatest risk for poorer pregnancy-related outcomes. Earlier identification and intervention through treatments targeting complement activation may decrease risk of developing adverse sequelae associated with severe COVID-19 disease in pregnancy.

In our cohort of pregnant participants with severe COVID-19, altered lipid metabolism, specifically changes in apolipoproteins and phosphatidylcholines, was identified, suggesting its role in the pathophysiology of development of severe disease. Prior multiomic studies of COVID-19 outside of pregnancy have also demonstrated similar associations with metabolic changes and alterations in lipid mediators. The downregulation of choline and its derivatives along with the dysregulation of various apolipoproteins were demonstrated to be more strongly associated with severe cases of COVID-19 compared to non-severe cases [13, 16]. Wu et al. similarly found decreased levels of phosphatidylcholines in fatal COVID-19 cases. In our cohort of pregnant participants with COVID-19, phosphatidylcholines and apolipoproteins were significantly downregulated in pregnancies with severe COVID-19, supporting the findings of alternative studies but specifically in the pregnant population [8]. The downregulation of apolipoproteins and phosphatidylcholines may be related to altered lipid metabolism, and there are previous reports that these biochemical differences have been observed prior to severe COVID-19 infection [19]. This dysregulation in lipid metabolism appears to play an important role in development of severe infection and based on our study findings may serve as an early marker for progression to severe disease. When including apolipoprotein A1 in a predictive model for predictor of severe COVID-19 disease with serum lbumin and CRP, we found that this composite measure performed better in identification of severe disease than the individual markers. Further studies should be performed to evaluate the validity of these

clinical lab parameters and determine whether they may serve a role as a predictive surrogate marker, allowing for earlier intervention and attempt to decrease progression of disease.

*Clinical implications.* Pregnant patients are often excluded from research studies. As such, there was a significant lack of knowledge on how COVID-19 effects the inflammatory and metabolic profile in pregnant patients. Our study shows that severe COVID-19 in pregnancy results in greater inflammation, complement activation and dysregulation of serum lipids, like the general population. As such, pregnant patients with COVID-19 should be managed similarly to their non-pregnant counterparts.

*Research implications.* We provide serum inflammatory and metabolic profiles in pregnant participants with COVID-19 using multiomic profiling with substantial implications for future mechanistic studies. Severe COVID was associated with dramatic proteomic and lipidomic changes, whereas Asymptomatic infections exhibited few significant changes compared to controls. Post-infection mild/moderate COVID-19 did not have significantly altered serum protein or compound levels compared to controls. Our study further elucidates the inflammatory and metabolic changes observed in severe COVID-19.

## Limitations

Limitations of this study include the relatively small sample sizes which may have prevented detection of smaller effect size associations. Only asymptomatic were collected during infection, mild and moderate cases were sample, resulting in lack of coverage of the effects of mild and moderate infection. By examining and analyzing other types of biospecimen, such as whole blood, plasma, or placental samples, we can further corroborate and provide robust data that supports our results. Higher BMI and associated comorbidities can also affect metabolomic profile making it challenging to differentiate which biochemicals changes are caused by obesity and which are caused by and predictive of severe COVID-19 [16]. There are opportunities for future studies to assess whether these signatures precede infection and might serve as risk biomarkers prior to actual infection. Furthermore, the specimens were collected in an era before vaccination was available and pre-Omicron. However, the strength is the untargeted screening of complete metabolites, proteins, and lipids for association with COVID-19 severity.

## Conclusions

To summarize, we find that there are dramatic lipidomic and proteomic changes that are associated with maternal COVID-19 severity. They appear to be associated with increased inflammation, e.g. immunoglobulins, acute phase response, SERPINs, and dysregulated lipid metabolism e.g. apolipoproteins, phosphatidylcholines, and acylglycerols, which have been observed in studies of COVID-19 severity in non-pregnant populations. Given the clear similarities observed in the metabolomic profiles and suggested underlying pathophysiology of COVID-19 infection in the pregnant and non-pregnant population, we emphasize the importance of caring for pregnant patients with COVID-19 with similar protocols as non-pregnant patients. The inclusion of pregnant people in mechanistic, therapeutic, and vaccine studies should be at the forefront of clinical considerations to improve health for pregnant people.

## Supporting information

**S1 Table. Differential gene expression by all modeling schema.**
(XLSX)

**S2 Table. Pathway enrichment.**
(XLSX)

**S1 Fig. Quality control.**
(PDF)

## Acknowledgments

We are indebted to the pregnant study participants for participation in our study.

Paper presentations: Findings from this paper were presented at the 41st annual Society for Maternal Fetal Medicine, Orlando, FL USA (January 31st–February 5th, 2022)

## Author Contributions

**Conceptualization:** Yalda Afshar.

**Data curation:** Thalia Mok, Christine Jang, Austin Quach, Yalda Afshar.

**Formal analysis:** Austin Quach, Yalda Afshar.

**Funding acquisition:** Marie Altendahl, Yalda Afshar.

**Investigation:** Marie Altendahl, Thalia Mok, Yalda Afshar.

**Methodology:** Marie Altendahl, Austin Quach, Yalda Afshar.

**Project administration:** Christine Jang, Yalda Afshar.

**Resources:** Seungjun Yeo, Austin Quach, Yalda Afshar.

**Software:** Austin Quach.

**Supervision:** Yalda Afshar.

**Validation:** Yalda Afshar.

**Visualization:** Marie Altendahl, Austin Quach, Yalda Afshar.

**Writing – original draft:** Marie Altendahl, Yalda Afshar.

**Writing – review & editing:** Marie Altendahl, Thalia Mok, Christine Jang, Seungjun Yeo, Austin Quach, Yalda Afshar.

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
