## [Decision Letter · Decision Letter 0]

29 Jul 2022

PONE-D-22-11635Severe COVID-19 in pregnancy has a distinct serum profile, including greater complement activation and dysregulation of serum lipidsPLOS ONE

Dear Dr. Afshar,

Thank you for submitting your manuscript to PLOS ONE. After careful consideration, we feel that it has merit but does not fully meet PLOS ONE’s publication criteria as it currently stands. Therefore, we invite you to submit a revised version of the manuscript that addresses the points raised during the review process. More specifically, you need to provide a more precise clinical picture of your patients concerning the severity of the disease. In addition,  the concentrations of antibodies against SARS-CoV-2 (IgM and IgG) as well as a cytokine profil need to be presented.

We look forward to receiving your revised manuscript.

Kind regards,

Catherine Mounier

Academic Editor

PLOS ONE

Journal Requirements:

“AQ and SY are co-founders of, are employed by, and own stock in Dalton Bioanalytics Inc. All other authors report no conflict of interest.”

Reviewers' comments:

Reviewer's Responses to Questions

**Comments to the Author**

1. Is the manuscript technically sound, and do the data support the conclusions?

Reviewer #1: Yes

Reviewer #2: Partly

2. Has the statistical analysis been performed appropriately and rigorously? 

Reviewer #1: Yes

Reviewer #2: N/A

3. Have the authors made all data underlying the findings in their manuscript fully available?

Reviewer #1: Yes

Reviewer #2: Yes

4. Is the manuscript presented in an intelligible fashion and written in standard English?

Reviewer #1: Yes

Reviewer #2: No

5. Review Comments to the Author

Reviewer #1: Severe COVID-19 in pregnancy has a distinct serum profile, including greater complement activation and dysregulation of serum lipids investigated the pathophysiology behind various clinical trajectories in pregnant patients diagnosed with COVID-19 using multi-omics profiling. The information contained in the manuscript is timely and makes a significant contribution to the body of literature related to the COVID-19 pandemic, particularly by presenting data from a less-studies population – pregnant persons. The manuscript is generally well-written; therefore, my comments/suggestions/questions are minimal.

Line 78: include (ARDS) after syndrome as this acronym is used later in the manuscript

Line 109: Healthy pregnant controls without COVID-19...

Line 114: moderate, or severe

Lines 125-126: This section could be revised for clarity. It’s a bit wordy and repetitious. Sentences should begin with numbers (i.e., 31 different serum samples). Were 2 samples collected? If so, why weren’t comparisons made? Were samples collected from more than 31 COVID-positive participants and the ones used here randomly selected? If so, how many and how was the selection made?

Line 152-158: verb tense changes and paragraph begins with a number.

Line 160: What do you mean by available subject variables? COVID / draw?

Line 169: A total of 30 participants… There are also several places where number format is incorrection in this section.

Line 177: COVID-19 severity. There are other areas of inconsistency with this term. Please double check

Line 189-190: I am not sure what this sentence is trying to say. It seems something is missing.

Line 236: I think this sentence is confusing, especially considering whether two samples were collected from participants and whether the samples were compared. If you only looked at one sample, the differences might be a better word than changes. Also, differences between pregnancies with and without COVID-19.

Table 1: There are 7 individuals in Race/Ethnicity for the Asymptomatic Group; Why is Gestational Age for 28 participants?

One last comment: Phosphatidylcholine concentrations change during pregnancy due to increased de novo synthesis in the maternal liver. In general, levels increase, particularly in the 2nd trimester, so trimester of measurement may matter. Additionally, infection, inflammation, etc. lead to sequestration of choline in the liver and alteration of the serum levels. I think acknowledging this in the discussion is important, especially as this study is specific to pregnant women and the Severe group was more likely to be obese.

Reviewer #2: The impact of COVID-19 on pregnancy is an important topic that needs close attention and thorough study. Thus the selection of the topic is of paramount importance. In the present study, the authors tried to evaluate the changes in the plasma proteins and lipids in those who are pregnant and having COVID-18 infection.

After measuring all the parameters, the authors concluded that Pregnancies with severe COVID-19 demonstrate greater inflammation and complement activation and dysregulation of serum lipids. This altered multiomic expression provides insight into the pathophysiology of severe COVID-19 in pregnancy - but did not provide information about cytokines profile, range of D-die=mer and ferritin levels.

Some of the issues are:

1. THe number of patients studied is small for a disease like COVID-19.

2. Definition of severity of COVID-19 need to be provided-such as clinical picture, oxygens aturation levels, cytokines levels, and other

3. In all the study population, authors could have measured concentrations of antibodies against SARS-CoV-2 -both IgM and IgG levels, how these titers correspond with the clinical picture and other clinical and lab parameters.

4. In a study of this type it is mandatory to measure plasma cytokines profile (TNF, IL6, MIF, IL-4 and IL-10) and correlate the same to the clinical picture, antibody titers and outcome of the infection.

5. If it is possible measurement of Treg and Teff cells would have been helpful.

Without these indices, the study becomes a routine clinical study with little insight into the disease pathobiology.

6. PLOS authors have the option to publish the peer review history of their article (what does this mean?). If published, this will include your full peer review and any attached files.

Reviewer #1: No

Reviewer #2: No

---

## [Author Response · Author response to Decision Letter 0]

22 Sep 2022

Point-by-Point Response: 

Journal Requirements:

1. Please ensure that your manuscript meets PLOS ONE's style requirements, including those for file naming. The PLOS ONE style templates can be found a https://journals.plos.org/plosone/s/file?id=wjVg/PLOSOne_formatting_sample_main_body.pdf

Thank you. Confirmed. 

We added more detail in the material and methods. “Pregnant people with COVID-19 infection were actively enrolled at the University of California, Los Angeles between October 1st, 2020 and February 28th, 2021 through a prospective cohort study evaluating maternal and neonatal outcomes of pregnancies with COVID-19 infection. The study was approved by the Institutional Review Board (UCLA IRB# 20-000579). After patient was identified, participants underwent screening and eligible participants were contacted by a study coordinator by phone or email. Verbal or email consented was obtained. Confirmed COVID-19 infection was defined as being SARS-CoV-2 positive by nasopharyngeal RT-PCR. Eligibility for the study included participants >13 years old, pregnant at the time of enrollment, with necessary clinical data and biospecimen collection during pregnancy. Healthy pregnant controls without COVID-19 infection, defined as a negative SARS-CoV-2 positive nasopharyngeal RT-PCR test, were concurrently recruited.”

“AQ and SY are co-founders of, are employed by, and own stock in Dalton Bioanalytics Inc. All other authors report no conflict of interest.” Please confirm that this does not alter your adherence to all PLOS ONE policies on sharing data and materials, by including the following statement: "This does not alter our adherence to PLOS ONE policies on sharing data and materials.” (as detailed online in our guide for authors http://journals.plos.org/plosone/s/competing-interests). If there are restrictions on sharing of data and/or materials, please state these. Please note that we cannot proceed with consideration of your article until this information has been declared. Please include your updated Competing Interests statement in your cover letter; we will change the online submission form on your behalf.

Thank you. We added: “we have no problems with sharing the data for this publication” on behalf of AQ and SY to the cover letter. 

Done.

Reviewers' comments:

Review Comments to the Author

Reviewer #1: Severe COVID-19 in pregnancy has a distinct serum profile, including greater complement activation and dysregulation of serum lipids investigated the pathophysiology behind various clinical trajectories in pregnant patients diagnosed with COVID-19 using multi-omics profiling. The information contained in the manuscript is timely and makes a significant contribution to the body of literature related to the COVID-19 pandemic, particularly by presenting data from a less-studies population – pregnant persons. The manuscript is generally well-written; therefore, my comments/suggestions/questions are minimal.

Thank you so much for the summary and the thoughtful comments and revisions throughout. We agree that pregnant persons are less studied, and we believe that with incorporation of your edits we have made the manuscript stronger. 

Line 78: include (ARDS) after syndrome as this acronym is used later in the manuscript

Done. Thank you. Updated: “The clinical presentation of Coronavirus Disease 2019 (COVID-19) in pregnancy is variable and can range in severity from asymptomatic to critical illness with acute respiratory distress syndrome (ARDS), multiorgan failure, and in some cases, death”

Line 109: Healthy pregnant controls without COVID-19...

Corrected. Updated: “Healthy pregnant controls without COVID-19 infection, defined as a negative SARS-CoV-2 positive nasopharyngeal RT-PCR test, were concurrently recruited.”

Line 114: moderate, or severe

Edited as recommended. Updated: “COVID-19 severity was categorized as asymptomatic, mild, moderate, or severe based on the NIH guidelines”

Lines 125-126: This section could be revised for clarity. It’s a bit wordy and repetitious. Sentences should begin with numbers (i.e., 31 different serum samples). Were 2 samples collected? If so, why weren’t comparisons made? Were samples collected from more than 31 COVID-positive participants and the ones used here randomly selected? If so, how many and how was the selection made?

Updated: “Peripheral blood specimens were obtained from consented patients at the time of study enrollment, as close to diagnosis of acute COVID-19 infection as possible… Of the 267 available samples, 31 serum samples were chosen based on COVID-19 severity (control, asymptomatic, mild/moderate, and severe) for analyses. Each sample was from a unique participant.”

Line 152-158: verb tense changes and paragraph begins with a number.

Thank you. We edited to improve the language. Updated: “Serum samples underwent multiomic profiling however a single outlier sample was omitted from further analysis likely due to mis-injection (technical error). Relative intensities were log10 transformed and statistically corrected for the technical effects run order and extraction efficiency (average internal standard intensity).”

Line 160: What do you mean by available subject variables? COVID / draw?

Updated: “Participant demographic and clinical characteristics were clustered based on pairwise correlation distance to identify potentially important associations.”

Line 169: A total of 30 participants… There are also several places where number format is incorrection in this section.

We are unsure what exactly this means but made sure on revision that all numbers consistent throughout the paper and we kept formatting consistent. We hope this helps ameliorate this issue.

Line 177: COVID-19 severity. There are other areas of inconsistency with this term. Please double check

Thank you for catching this consistency. Updated all “COVID” to COVID-19

Line 189-190: I am not sure what this sentence is trying to say. It seems something is missing.

“Precipitous changes in maternal serum were seen in those with severe COVID-19 infection in pregnancy.”

Line 236: I think this sentence is confusing, especially considering whether two samples were collected from participants and whether the samples were compared. If you only looked at one sample, the differences might be a better word than changes. Also, differences between pregnancies with and without COVID-19.

Updated: “The purpose of this study was to identify metabolic and proteomic differences seen in pregnancies with and without COVID-19 and evaluate for metabolic and proteomic differences between severity of COVID-19 infection. “

Table 1: There are 7 individuals in Race/Ethnicity for the Asymptomatic Group; Why is Gestational Age for 28 participants?

Updated table, “2 participants did not deliver within the University of California, Los Angeles Health system and thus were missing birthing outcome variables such as gestational age at delivery.”

One last comment: Phosphatidylcholine concentrations change during pregnancy due to increased de novo synthesis in the maternal liver. In general, levels increase, particularly in the 2nd trimester, so trimester of measurement may matter. Additionally, infection, inflammation, etc. lead to sequestration of choline in the liver and alteration of the serum levels. I think acknowledging this in the discussion is important, especially as this study is specific to pregnant women and the Severe group was more likely to be obese.

Thank you for this important clarifying point and comment. We totally agree and appreciate the gestational age changes as well as the BMI correlation and as such we control for both of these parameters in the models we present. This is a very important clinically relevant modifier and hence built into our model. You will see that new models that are included consider these changes and are detailed below: 

lm1: severeCOVID

lm1b: severeActiveCOVID

lm2a: severeCOVID + age + race

lm2b: severeCOVID + age + race + covid_ga_w + BMI_delivery

lm2c: severeCOVID + age + race + covid_ga_w + med_cond___obesity

lm3: covid_severity_recent

lm4: covid_severity_recent [excluding severe cases] {basically asymptomatic vs non- and post-infection controls}

lm5: covid_status [excluding COVID within 6 weeks of sampling] {basically post-infection mild/moderate vs controls}

Reviewer #2: The impact of COVID-19 on pregnancy is an important topic that needs close attention and thorough study. Thus the selection of the topic is of paramount importance. In the present study, the authors tried to evaluate the changes in the plasma proteins and lipids in those who are pregnant and having COVID-18 infection.

Thank you. We agree that pregnant people are understudied and we are thrilled to use this technology, LC-MS, metabolomics, proteomics, and lipidomics, in this population which would supplement the vast work on cytokines and antibody response in pregnant patients.

After measuring all the parameters, the authors concluded that Pregnancies with severe COVID-19 demonstrate greater inflammation and complement activation and dysregulation of serum lipids. This altered multiomic expression provides insight into the pathophysiology of severe COVID-19 in pregnancy - but did not provide information about cytokines profile, range of D-dimer and ferritin levels.

Our focus is on metabolomics, proteomics, and lipidomics. We discuss below that cytokine measurement is not possible by LC-MS which was the focus of this paper. The measurement of cytokines is generally well below the limits of detection for untargeted, undepleted plasma proteomics experiments (~nanomolar) due to low concentration (~picomolar) and interference of high abundance species (e.g. serum albumin). Even with specialized affinity immuno-capture techniques, limits of detection are in the high picomolar range [Stenken, J. A., & Poschenrieder, A. J. (2015). Bioanalytical chemistry of cytokines–a review. Analytica chimica acta, 853, 95-115.]. Thus, alternative approaches (e.g. immunoassays) are better suited for cytokine measurement. The targets of interest to the reviewer were not detected (ferritin, TNF, IL6, , MIF, IL4, IL10, D-dimer) likely because they are too low in concentration (~picomolar). We did measure fibrinogen chains A, B and G (coagulation/D-dimer related) but these were not associated with COVID-19 status in our study. All this data is available in Supplemental Table 1.

Some of the issues are:

1. THe number of patients studied is small for a disease like COVID-19.

We agree that having more patients would be great and our cohort has grown significantly. However, to be able to do a single run of LC-MS for each sample costs $2,500 per sample (just for running the sample, no analysis, and no personnel). The cost of the experimental run for this paper was close to $75,000 USD. We believe work like ours with deep phenotyping in Supplemental Tables and clinical correlates (Table 1-2) that we provide are ripe for other researchers to continue to ask more focused questions after we provide an -omics tools here for analysis. 

2. Definition of severity of COVID-19 need to be provided-such as clinical picture, oxygens aturation levels, cytokines levels, and other

COVID-19 severity is standardized by the US FDA and IDSA, among others. We use the standardized definitions throughout. Tables 1 and 2 include details of the clinical characteristics. 

3. In all the study population, authors could have measured concentrations of antibodies against SARS-CoV-2 -both IgM and IgG levels, how these titers correspond with the clinical picture and other clinical and lab parameters.

Briefly here and answered in more detail below. IgG and IgM constant heavy chains are found in the data; however, we realize that the reviewer is likely looking for g/L absolute quant and we only have relative intensity, which is what is done with LC-MS The relative quant for those heavy constant chains are already reported in the results table. The objective here was not related to antibody responses and focus on proteomics, metabolomics, and lipidomics. 

4. In a study of this type it is mandatory to measure plasma cytokines profile (TNF, IL6, MIF, IL-4 and IL-10) and correlate the same to the clinical picture, antibody titers and outcome of the infection.

To answer both comment #3 and #4 which recommend the study of antibodies (#3) and cytokines (#4) these are very interesting molecular mechanistic questions that many groups (including us) are tackling and have published on (see below work that our group has done on antibodies and cytokines):

Our publications on this topic

Cambou MC, , et al. J Infect Dis. 2022 Sep 9. PMID: 36082433

Matsui Y, et al. JCI Insight. 2022 Jun 22;7(12): PMID: 35579965. 

Foo SS, et al. Cell Rep Med. 2021 Nov 16; PMID: 34723226

Collaborators/others publications on this topic

Atyeo C, et al Cell. 2021 PMID: 33476549; PMCID: PMC7755577.

Flannery DD, et al. JAMA Pediatr. 2021 Jun 1. PMID: 33512440; PMCID: PMC7846944.

Edlow AG, et al. JAMA Netw Open. 2020 Dec 1: 33351086; PMCID: PMC7756241.

However, the objective in this paper and in our paper was the novelty that it is not related to the SARS-CoV2 antibody response, as we have surrogate of immunoglobulin already measured in our .xls presented but this is NOT specific to SARS CoV2 and we did not feel was relevant, rather we use a novel technology and methodology capable of profiling proteins, lipids, electrolytes, metabolites, drugs, environmental chemicals, and other molecules in a single assay. 

We agree that plasma cytokines and COVID-19 profiles are interesting as they correlate to clinical picture and outcomes. Our study team has investigated this and some work has been published here (Foo SS, Cambou MC, Mok T, et al. Cell Rep Med. 2021 Nov 16;2(11):100453. 2021 Oct 27. PMID: 34723226; PMCID: PMC8549189.). However, our objective and methodology are not focused on cytokine evaluation as this requires a very different methodology and here we are interested in something aside from cytokines that have been well studied using proteomic, lipidomics, and metabolomics. The measurement of cytokines is generally well below the limits of detection for untargeted, undepleted plasma proteomics experiments (~nanomolar) due to low concentration (~picomolar) and interference of high abundance species (e.g. serum albumin). Even with specialized affinity immuno-capture techniques, limits of detection are in the high picomolar range [Stenken, J. A., & Poschenrieder, A. J. (2015). Bioanalytical chemistry of cytokines–a review. Analytica chimica acta, 853, 95-115.]. Thus, alternative approaches (e.g. immunoassays) are better suited for cytokine measurement.

5. If it is possible measurement of Treg and Teff cells would have been helpful.

The study of immune populations of Treg and Teff cells are interesting, but not the focus of this work and actually require a different source material than we used. They require the peripheral mononuclear cells (PBMCs) from heparinized the whole blood isolated, versus the heparinized blood (serum) used in our experimental design to ask what the changes at the level of the proteins, metabolomics, and lipids are The laboratories of Drs. Nardy-Gomez, Franco, Collier, and many others have published on this population in pregnant populations with COVID-1

Hsieh LE, et al. J Reprod Immunol. 2022 Feb;149:103464. doi: 10.1016/j.jri.2021.103464. Epub 2021 Dec 11. PMID: 34953325; PMCID: PMC8665650.

Garcia-Flores V, et al. Nat Commun. 2022 Jan 18;13(1):320. doi: 10.1038/s41467-021-27745-z. PMID: 35042863; PMCID: PMC8766450.

---

## [Decision Letter · Decision Letter 1]

13 Oct 2022

Severe COVID-19 in pregnancy has a distinct serum profile, including greater complement activation and dysregulation of serum lipids

PONE-D-22-11635R1

Dear Dr. Afshar,

We’re pleased to inform you that your manuscript has been judged scientifically suitable for publication and will be formally accepted for publication once it meets all outstanding technical requirements.

Kind regards,

Catherine Mounier

Academic Editor

PLOS ONE

Additional Editor Comments (optional):

Reviewers' comments:

Reviewer's Responses to Questions

**Comments to the Author**

1. If the authors have adequately addressed your comments raised in a previous round of review and you feel that this manuscript is now acceptable for publication, you may indicate that here to bypass the “Comments to the Author” section, enter your conflict of interest statement in the “Confidential to Editor” section, and submit your "Accept" recommendation.

Reviewer #2: All comments have been addressed

2. Is the manuscript technically sound, and do the data support the conclusions?

Reviewer #2: Yes

3. Has the statistical analysis been performed appropriately and rigorously? 

Reviewer #2: Yes

4. Have the authors made all data underlying the findings in their manuscript fully available?

Reviewer #2: Yes

5. Is the manuscript presented in an intelligible fashion and written in standard English?

Reviewer #2: Yes

6. Review Comments to the Author

Reviewer #2: Authors have addressed majority of the questions.

THe only surprise is that cytokines data is not available that would have been interesting. Agreed that others have published this data but it would be interesting if cytokines data is correlated with the other measuremetns done by the authors.

7. PLOS authors have the option to publish the peer review history of their article (what does this mean?). If published, this will include your full peer review and any attached files.

Reviewer #2: **Yes: **Undurti N Das

---

## [Editor Report · Acceptance letter]

20 Oct 2022

PONE-D-22-11635R1 

Severe COVID-19 in pregnancy has a distinct serum profile, including greater complement activation and dysregulation of serum lipids 

Dear Dr. Afshar:

I'm pleased to inform you that your manuscript has been deemed suitable for publication in PLOS ONE. Congratulations! Your manuscript is now with our production department. 

Kind regards, 

on behalf of

Dr. Catherine Mounier 

Academic Editor

PLOS ONE